



# Tropospheric ozone changes and ozone sensitivity from present-day to future under shared socio-economic pathways

Zhenze Liu[1], Ruth M. Doherty[1], Oliver Wild[2], Fiona M. O'Connor[3], and Steven T. Turnock[3,4]

[1]School of GeoSciences, The University of Edinburgh, Edinburgh, UK
[2]Lancaster Environment Centre, Lancaster University, Lancaster, UK
[3]Met Office Hadley Centre, Exeter, UK
[4]University of Leeds Met Office Strategic Research Group, School of Earth and Environment, University of Leeds, Leeds, UK

**Correspondence:** Zhenze Liu (zhenze.liu@ed.ac.uk)

**Abstract.** Tropospheric ozone is important to future air quality and climate. We investigate ozone changes and ozone sensitivity to changing emissions in the context of climate change from the present day (2004–2014) to the future (2045–2055) under a range of shared socio-economic pathways (SSPs). We apply the United Kingdom Earth System Model, UKESM1, with an extended chemistry scheme including more reactive volatile organic compounds (VOCs) to quantify ozone burdens as well as ozone sensitivities globally and regionally based on nitrogen oxide ($NO_x$) and VOC concentrations. We show that the tropospheric ozone burden increases by 4 % under a development pathway with higher $NO_x$ and VOC emissions (SSP3-7.0), but decreases by 7 % under the same pathway if $NO_x$ and VOC emissions are reduced (SSP3-7.0-lowNTCF) and by 5 % if atmospheric methane ($CH_4$) concentrations are reduced (SSP3-7.0-lowCH4). Global mean surface ozone concentrations are reduced by 3–5 ppb under SSP3-7.0-lowNTCF and by 2–3 ppb under SSP3-7.0-lowCH4. However, surface ozone changes vary substantially by season in high-emission regions under future pathways, with decreased ozone concentrations in summer and increased ozone concentrations in winter when $NO_x$ emissions are reduced. VOC-limited areas are more extensive in winter (7 %) than in summer (3 %) across the globe. North America, Europe and East Asia are the dominant VOC-limited regions in the present day but North America and Europe become more $NO_x$-limited in the future mainly due to reductions in $NO_x$ emissions. The impacts of VOC emissions on ozone sensitivity are limited in North America and Europe because reduced anthropogenic VOC emissions are offset by higher biogenic VOC emissions. Ozone sensitivity is not greatly influenced by changing $CH_4$ concentrations. South Asia becomes the dominant VOC-limited region under future pathways. We highlight that reductions in $NO_x$ emissions are required to transform ozone production from VOC- to $NO_x$-limitation, but that these lead to increased ozone concentrations in high-emission regions, and hence emission controls on VOC and $CH_4$ are also necessary.

## 1 Introduction

Ozone ($O_3$) is a chemically reactive component in the atmosphere that is produced from natural and anthropogenic sources. Emissions of $O_3$ precursors including nitrogen oxides ($NO_x$), volatile organic compounds (VOCs), methane ($CH_4$) and carbon monoxide (CO) lead to the formation of $O_3$ by a series of photochemical reactions in the presence of sunlight. $O_3$ has important impacts on human health, ecosystems and climate change (Lefohn et al., 2018; Zhang et al., 2019; Agathokleous et al., 2020).





$O_3$ concentrations are largely governed by the magnitudes of $O_3$ precursor emissions, transport, deposition and transport from
the stratosphere. $O_3$ exerts a positive radiative forcing on climate forcing (Stevenson et al., 2013; O'Connor et al., 2021;
Thornhill et al., 2021a), and changes in climate in turn influence ozone (Fiore et al., 2012; Doherty et al., 2013). Climate
change can alter natural emissions of biogenic VOCs (BVOC), lightning $NO_x$ and $CH_4$, along with temperature, humidity,
convection and clouds, which further influence $O_3$ concentrations (Thornhill et al., 2021b). The interactions between air quality
and climate play an important role in the coupled Earth system, and we focus on the impacts of future emissions in the context
of climate change on tropospheric $O_3$ in this study.

The tropospheric $O_3$ burden is controlled by the amount of $O_3$ production, $O_3$ destruction, $O_3$ deposition and the $O_3$
transport from the stratosphere (Lelieveld and Dentener, 2000; Wild, 2007). From pre-industrial times to the present day, the
tropospheric $O_3$ burden has increased from approximately 240 Tg to 350 Tg mainly due to substantial increases in anthro-
pogenic $O_3$ precursor emissions (Lamarque et al., 2010; Young et al., 2013; Griffiths et al., 2021). However, regional surface
$O_3$ changes between the pre-industrial and present day vary substantially due to different regional emission changes (Turnock
et al., 2020) and to differences in $O_3$ sensitivity to $NO_x$ and VOC emissions. In recent decades, there has been a decrease in
surface $O_3$ concentrations in North America and Europe due to emission controls (Lefohn et al., 2008; Colette et al., 2016).
In contrast, increases in surface $O_3$ levels are observed in South Asia and East Asia due to industrialization, urbanization and
social development (Akimoto, 2003; Ohara et al., 2007). Furthermore, while emission controls have been implemented across
industrial regions of China in recent years, these have focused on emissions of $NO_x$ and particulate matter, and have led to
increased $O_3$ pollution in some places (Wang et al., 2017; Silver et al., 2018).

It is important to investigate $O_3$ sensitivity to understand how $O_3$ chemical regimes might change in different parts of the
world, and to guide suitable emission control strategies. However, few studies have focused on $O_3$ sensitivity from a global
perspective, or on how this might change in the future. $O_3$ sensitivity, typically characterized by a $NO_x$- or VOC-limited
$O_3$ production regime, is dependent on the relative abundances of $NO_x$ and VOC concentrations (Sillman, 1995, 1999), and
determines the extent and effectiveness of different emission control strategies. VOC-limited regimes typically occur in highly
urbanised regions with high $NO_x$ concentrations in which decreases in $NO_x$ emissions increase $O_3$ concentrations, and $O_3$
production increases with higher VOC emissions. In contrast, changes in $NO_x$ concentrations dominate $O_3$ changes in $NO_x$-
limited regimes such that decreases in $NO_x$ emissions decrease $O_3$ concentrations, and $O_3$ concentrations are less sensitive to
VOC emissions. $O_3$ sensitivity can be assessed by quantifying the ratio between $NO_x$ and VOC concentrations and we apply
this approach in this study to present-day and future conditions.

The shared socio-economic pathways (SSPs) are future emission and climate scenarios accounting for future social, eco-
nomic and environmental developments (O'Neill et al., 2014; van Vuuren et al., 2014). The SSPs represent a range of levels of
policy strength (weak, medium and strong) to control emissions of near-term climate forcers (NTCFs) that include tropospheric
$O_3$, $O_3$ precursors and aerosols (Rao et al., 2017). Our study is based on simulations using historical and future SSPs emis-
sions and climate undertaken as part of the Aerosol Chemistry Model Intercomparison Project (AerChemMIP; Collins et al.,
2017) and the wider Coupled-Model Intercomparison Project Phase 6 (CMIP6; Eyring et al., 2016). The aim of AerChemMIP



is to quantify the effects of chemistry and aerosols on air quality and climate in CMIP6 by conducting historical and future experiments using chemistry-climate models with specified climate and emission trajectories.

We examine tropospheric $O_3$ and surface $O_3$ sensitivity under present-day (2004–2014) and future conditions (2045–2055). Model development and application are described in Sect. 2 along with descriptions of the emission and climate scenarios used. We compare and evaluate the present-day tropospheric $O_3$ burden and surface $O_3$ concentrations with two different chemistry schemes in Sect. 3. We then investigate the seasonal, daytime and nighttime differences in $O_3$ changes in the future compared to present-day for different regions in Sect. 4. Analysis of $O_3$ concentrations and production is used to quantify $O_3$ sensitivity

and to explain contrasting regional $O_3$ changes in Sect. 5. We then show the changes in $O_3$ sensitivity between different seasons and scenarios in Sect. 6 and present our conclusions in Sect. 7.

## 2    Materials and methods

### 2.1    Model description, development and application

We use version 1 of the United Kingdom Earth System Model, UKESM1 (Sellar et al., 2019), to reproduce present-day

(2004–2014) $O_3$ concentrations and to predict $O_3$ responses to emissions and climate in the future (2045–2055). UKESM1 consists of a physical climate model, the Hadley Centre Global Environment Model version 3 (HadGEM-GC3.1), configured with the Global Atmosphere 7.1 and Global Land 7.0 (GA7.1/GL7.0) components (Walters et al., 2019) to which Earth System (ES) processes have been coupled (Sellar et al., 2019). Atmospheric composition is modelled using a state-of-the-art chemistry and aerosol module, the United Kingdom Chemistry and Aerosol model (UKCA; Morgenstern et al., 2009; O'Connor et al.,

2014). UKCA includes a stratosphere-troposphere gas-phase chemistry scheme (StratTrop; Archibald et al., 2020a) coupled to the aerosol scheme GLOMAP-mode (Mann et al., 2010; Mulcahy et al., 2020). The model resolution is N96L85 in the atmosphere, with 1.875° in longitude by 1.25° in latitude, 85 terrain-following hybrid height layers and a model top at 85 km.

While the UKESM1 configuration for CMIP6 used the UKCA StratTrop mechanism, this study also uses an extended gas-phase chemistry scheme that incorporates more reactive VOC species to permit a more realistic representation of $O_3$ production

in polluted environments. The extended chemistry scheme (denoted as Ext_StratTrop hereafter) is based on the StratTrop scheme and includes oxidation of the additional chemical components propene ($C_3H_6$), butane ($C_4H_{10}$) and toluene ($C_7H_8$) to represent alkenes, alkanes and aromatic VOC classes, as described in Liu et al. (2021). The extended chemistry scheme includes 101 species, 244 bimolecular reactions, 26 uni- and termolecular reactions, 70 photolytic reactions, 5 heterogeneous reactions and 3 aqueous phase reactions for the sulphur cycle.

The atmosphere-only configuration of UKESM1 is used with prescribed sea surface temperatures and sea ice to show the transient impacts of emissions under present-day and future climates. These are prescribed using monthly-mean time-evolving fields from the fully coupled UKESM1. Greenhouse gas concentrations are prescribed as in historical and future simulations conducted by UKESM1 as part of CMIP6 (Meinshausen et al., 2017, 2020).



## 2.2 Emissions and experiments

Present-day CMIP6 anthropogenic and biomass burning emissions are taken from Hoesly et al. (2018) and van Marle et al. (2017), respectively. Biogenic VOC emissions are calculated interactively within the iBVOC emissions scheme (Pacifico et al., 2011) in the Joint UK Land Environmental Simulator (JULES) land-surface scheme which is coupled to UKCA. Other aspects of the emissions used here are the same as described in Turnock et al. (2020). Anthropogenic emissions are categorised into five sectors (industry, power plants, transport, residences and agriculture) as inputs to the model, with independent diurnal and 95 vertical emission profiles applied for each sector (Bieser et al., 2011; Mailler et al., 2013; Liu et al., 2021).

Three CMIP6 SSP scenarios are used for future simulations: SSP3-7.0, SSP3-7.0-lowNTCF and SSP3-7.0-lowCH4. SSP3-7.0 is a 'regional rivalry' policy scenario with a large anthropogenic climate forcing signal (a radiative forcing of 7.0 W m-2 at 2100). It has weak emission controls on $O_3$ precursors and aerosols, and rapidly increasing $CH_4$ concentrations (Fujimori et al., 2017; Rao et al., 2017). SSP3-7.0-lowNTCF and SSP3-7.0-lowCH4 are additional pathways which use the same underlying 100 climate policies as SSP3-7.0. SSP3-7.0-lowNTCF has strong controls on all NTCF emissions. SSP3-7.0-lowCH4 follows SSP3-7.0 but assumes strong mitigation of $CH_4$ emissions. BVOC emissions increase under all these pathways due to a warmer climate. We perform four model experiments in this study to investigate tropospheric $O_3$ for the present-day (2004–2014) and three future pathways (SSP3-7.0, SSP3-7.0-lowNTCF and SSP3-7.0-lowCH4; 2045–2055). Table 1 shows the model configuration for the four simulations. Table 2 lists the CMIP6 global mean NTCF total surface emissions and surface $CH_4$ 105 concentrations for the four scenarios.

**Table 1.** Model configurations for present-day and future simulations. "Emissions" refers to emissions of $O_3$ precursors and aerosols. "$CH_4$ conc." refers to prescribed surface $CH_4$ concentrations. "SST/SI" refers to prescribed sea surface temperature and sea ice concentrations. "Historical" means that the emissions, $CH_4$ concentrations or SST/SI evolve as for the CMIP6 historical simulations, and "Reference" means that they evolve as for SSP3-7.0. "Low" emissions or $CH_4$ concentrations evolve following SSP3-7.0 but with lower emissions or $CH_4$ concentrations.

| Experiment name | Time period | Emissions | $CH_4$ conc. | SST/SI |
|---|---|---|---|---|
| Present day | 2004–2014 | Historical | Historical | Historical |
| SSP370 | 2045–2055 | Reference | Reference | Reference |
| SSP370_lowNTCF | 2045–2055 | Low | Reference | Reference |
| SSP370_lowCH4 | 2045–2055 | Reference | Low | Reference |



**Table 2.** Overview of global annual mean time-varying surface emissions of $NO_x$, VOCs, CO, sulfur dioxide ($SO_2$), black carbon (BC) and organic carbon (OC) from anthropogenic (ANT), biomass burning (BB), biogenic (BIO) sources for the present day (2004–2014) and future (2045–2055) SSP3-7.0, SSP3-7.0-lowNTCF and SSP3-7.0-lowCH4. Annual mean surface $CH_4$ concentrations (ppb) are also shown.

| Emission (Tg yr$^{-1}$) | | Present day | SSP3-7.0 | SSP3-7.0-lowNTCF | SSP3-7.0-lowCH4 |
|---|---|---|---|---|---|
| $NO_x$ | ANT | 136.0 | 149.6 | 68.8 | 149.6 |
| | BB | 13.6 | 12.1 | 10.2 | 12.1 |
| | Total | 149.6 | 161.7 | 79.0 | 161.7 |
| VOCs | ANT | 156.3 | 195.3 | 117.6 | 195.3 |
| | BB | 62.7 | 57.1 | 47.5 | 57.1 |
| | BIO | 727.9 | 786.1 | 795.2 | 785.7 |
| | Total | 946.9 | 1038.5 | 960.3 | 1038.1 |
| CO | ANT | 600.8 | 662.7 | 328.4 | 662.7 |
| | BB | 324.6 | 318.5 | 264.5 | 318.5 |
| | Total | 925.4 | 981.2 | 592.9 | 981.2 |
| $SO_2$ | ANT | 115.4 | 95.7 | 43.0 | 95.7 |
| | BB | 2.1 | 2.2 | 1.8 | 2.2 |
| | Total | 117.5 | 97.9 | 44.8 | 97.9 |
| BC | ANT | 7.4 | 9.1 | 4.4 | 9.1 |
| | BB | 1.7 | 1.7 | 1.4 | 1.7 |
| | Total | 9.1 | 10.8 | 5.8 | 10.8 |
| OC | ANT | 18.0 | 23.3 | 10.1 | 23.3 |
| | BB | 15.0 | 14.5 | 11.9 | 14.5 |
| | Total | 33.0 | 37.8 | 22.0 | 37.8 |
| $CH_4$ (ppb) | | 1802.8 | 2471.9 | 2471.9 | 1363.7 |

# 3 Model evaluation of tropospheric and surface $O_3$

## 3.1 Comparison of StratTrop and extended chemistry schemes

We first compare averaged tropospheric $O_3$ burdens, chemical lifetime, chemical production, chemical loss and deposition during 2004–2014 from the extended chemistry scheme (Ext_StratTrop) with those from the StratTrop chemistry scheme used in AerChemMIP simulations (Table 3). We define the $O_3$ production rate as the sum of reactions fluxes through $HO_2/RO_2$ + NO, and the $O_3$ loss rate as the sum of $O(^1D) + H_2O$, $O_3 + HO_2/OH$/alkenes. The $O_3$ burden with the extended chemistry scheme (376 Tg) lies at the upper end of the uncertainty range for the observed burden, $340 \pm 34$ Tg for 2000 (Archibald



et al., 2020b). The magnitude of the $O_3$ burden is also consistent with the CMIP6 multi-model mean burden of $356 \pm 31$ Tg for 2005–2014 (Griffiths et al., 2021). The extended chemistry scheme produces a 5 % higher tropospheric $O_3$ burden than

that of StratTrop (358 Tg), demonstrating a more reactive environment for net $O_3$ formation throughout the troposphere due to reactive VOCs. This is also reflected in the higher rates of chemical $O_3$ production (11 %), loss (6 %) and deposition (6 %) with the extended chemistry scheme. However, the higher $O_3$ production is offset by greater $O_3$ destruction and by faster $O_3$ deposition, and hence the mean $O_3$ chemical lifetime remains very similar at about 22 days, which is consistent with previous multi-model estimates of the mean lifetime of $22.2 \pm 2.2$ days (Stevenson et al., 2006).

**Table 3.** Comparison of tropospheric $O_3$ burden and budget terms during 2004–2014 simulated with StratTrop and Ext_StratTrop chemistry schemes in UKCA. One standard deviation in the annual terms over the 2004–2014 period is shown. We define the tropopause based on the highest layer with an $O_3$ concentration less than 150 ppb.

|  | StratTrop | Ext_StratTrop |
| --- | --- | --- |
| $O_3$ burden (Tg) | 358±3 | 376±3 |
| $O_3$ lifetime (days) | 22.6±0.2 | 22.5±0.2 |
| $O_3$ net production (Tg year$^{-1}$) | 895±45 | 996±40 |
| $O_3$ production (Tg year$^{-1}$) | 5698±40 | 6080±66 |
| $O_3$ loss (Tg year$^{-1}$) | 4803±45 | 5084±59 |
| $O_3$ deposition (Tg year$^{-1}$) | 883±9 | 936±9 |

Simulated surface $O_3$ concentrations during 2004–2014 with the two chemistry schemes are compared in Fig. 1. Using the extended chemistry scheme, the spatial distribution of surface $O_3$ is similar to that using StratTrop in both winter and summer, but shows a general increase in global $O_3$ levels of about 2 ppb (Fig. 1a–d) due to inclusion of the additional reactive VOCs. The $O_3$ increases (Fig. 1e–f) are most notable for South Asia and East Asia due to the relatively high VOC emissions in these regions (Janssens-Maenhout et al., 2015; Huang et al., 2017; Feng et al., 2020). There is a much larger $O_3$ increase in South

Asia in winter than in summer (Fig. 1e), mainly due to greater transport of $O_3$ precursors during the summer monsoon in South Asia (Lu et al., 2018). This leads to higher $O_3$ concentrations in winter than that in summer, consistent with Gao et al. (2020), and demonstrates a larger seasonal variation in $O_3$ concentrations with the Ext_StratTrop chemistry scheme than with the StratTrop chemistry scheme. In addition, there are substantial $O_3$ increases in East Asia and in other polluted continents in summer (Fig. 1f) when using the extended chemistry scheme. This is not seen in winter (Fig. 1e) because titration of $O_3$ by

nitric oxide (NO) remains strong, despite high VOC emissions. The relatively small influence of the additional VOCs on $O_3$ in winter is also seen in other heavily populated regions that have high $NO_x$ emissions such as North America and Europe.





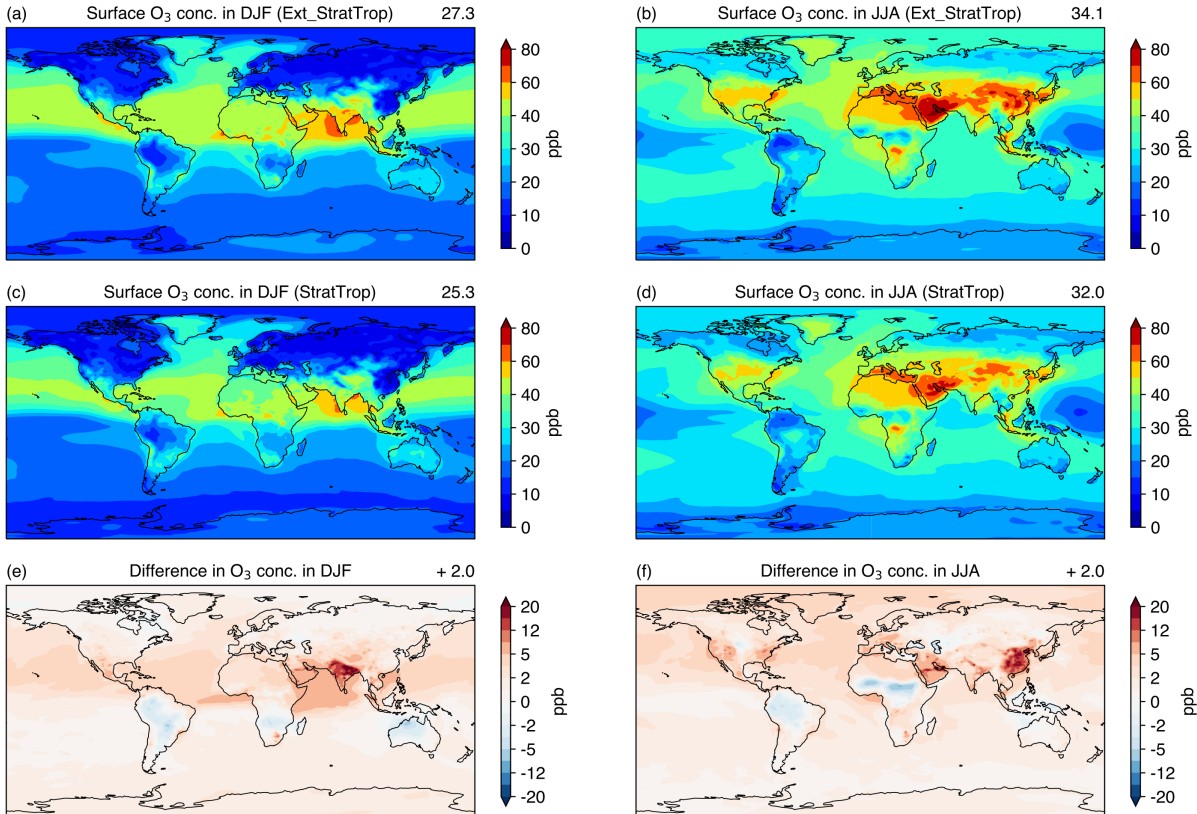

**Figure 1.** Comparison of present-day seasonal mean surface $O_3$ concentrations (2004–2014) between the Ext_StratTrop chemistry scheme (**a, b**) and the StratTrop chemistry scheme (**c, d**). Winter time (December–January–February, DJF) and summer time (June–July–August, JJA) global mean $O_3$ concentrations are shown. Seasonal differences in surface $O_3$ concentrations between the two chemistry schemes are shown in the bottom panels (**e, f**).

## 3.2 Evaluation of surface $O_3$ concentrations

We now evaluate surface $O_3$ concentrations simulated with the Ext_StratTrop chemistry scheme against gridded monthly mean rural observations from the TOAR dataset over the 2004–2014 period (Schultz et al., 2017). Surface $O_3$ concentrations

for winter (DJF) and summer (JJA) during 2004–2014 are shown in Fig. 2. We find that global mean surface $O_3$ concentrations are underestimated in winter (-7.3 ppb) and overestimated in summer (+13.5 ppb), with relatively small biases in spring and autumn. The positive biases in summer and negative biases in winter for $O_3$ concentrations are also seen in results using the StratTrop chemistry scheme (Archibald et al., 2020a; Turnock et al., 2020). The model seasonality with both schemes (19.4–45.5 ppb; DJF-JJA) is rather stronger than that observed (26.7–32 ppb; DJF-JJA), but the Ext_StratTrop chemistry

scheme improves the model performance slightly in DJF for surface $O_3$ (Fig. 2b) despite larger biases in summer (Fig. 2d). Numerical diffusion of $O_3$ precursor emissions due to coarse model horizontal resolution may explain the biases (Wild and





Prather, 2006; Stock et al., 2014; Fenech et al., 2018), and we note that these seasonal $O_3$ biases are also evident in other chemistry-climate models (Young et al., 2018; Turnock et al., 2020). Insufficient turbulent mixing in the planetary boundary layer may also contribute to the bias (O'Connor et al., 2014), as accumulation of $NO_x$ at the surface leads to greater $O_3$

production in summer and greater titration by NO in winter. However, we note that a much more comprehensive chemistry scheme applied in UKCA, the Common Representative Intermediates Mechanism (CRI-Strat), shows similar systematic biases in surface $O_3$ concentrations, -4.6 ppb in DJF and +12.0 ppb in JJA for the 2010–2018 period (Archer-Nicholls et al., 2020). This suggests that other biases in the model are primarily responsible for the biases in surface $O_3$. We choose to apply the extended chemistry scheme as it permits representation of a more appropriate chemical environment for $O_3$ production in

high-emission areas and is thus more suitable to investigate $O_3$ sensitivity.

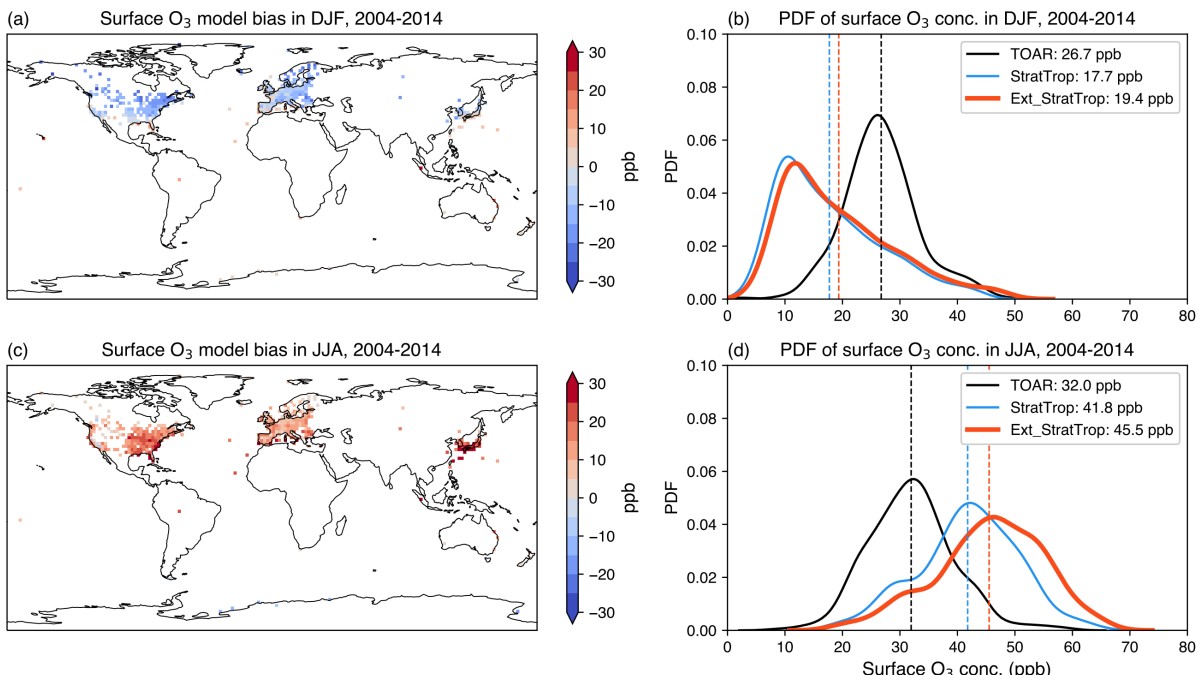

**Figure 2.** Modelled surface $O_3$ biases with the Ext_StratTrop chemistry scheme for winter (DJF) and summer (JJA) over 2004–2014 **(a, c)**. Probability distribution function (PDF) of seasonal mean $O_3$ concentrations between observations, StratTrop and Ext_StratTrop chemistry schemes **(b, d)**. Vertical dashed lines indicate seasonal mean surface $O_3$ concentrations. Observations from the Tropospheric Ozone Assessment Report (TOAR) dataset (Schultz et al., 2017) are used for comparison.



## 4   O$_3$ changes under future scenarios

### 4.1   Emission changes

The SSP3-7.0 pathway is characterized by relatively strong emission controls in some parts of the world such as North America and Europe but weaker controls or emission increases elsewhere. Increases in NO$_x$ and VOC emissions from anthropogenic and biomass burning sources are seen in Central and South America, North Africa, the Middle East, and South and East Asia (Fig. 3a, c). SSP3-7.0-lowCH4 has the same NTCF emissions as SSP3-7.0 but lower CH$_4$ emissions that lead to lower CH$_4$ concentrations (Table 2). SSP3-7.0-lowNTCF represents strong emission controls across the globe, with reductions in emissions in most major high-emission regions except for South Asia (Fig. 3b, d). Total BVOC emission changes are driven by changes in land-use, vegetation and temperature. Fig. 3e, f shows general increases in total BVOC emissions in most parts of the Northern Hemisphere except for South Asia in the future, which offset the decreased anthropogenic and biomass burning VOC emissions in North America and Europe. Similar BVOC emission changes are found under SSP3-7.0 and SSP3-7.0-lowNTCF relative to the present day as the climate change signal and carbon dioxide concentrations under future pathways are the same.

### 4.2   Tropospheric O$_3$ changes

Changes in tropospheric O$_3$ from the present day to the future are shown in Table 4. Changes in NTCF emissions control O$_3$ burden changes with a 4 % increase in the O$_3$ burden under SSP3-7.0 and a 7 % decrease under SSP3-7.0-lowNTCF relative to the present day. Changes in O$_3$ production rates are also controlled by changes in NTCF emissions, and there is higher O$_3$ production under SSP3-7.0 and lower O$_3$ production under SSP3-7.0-lowNTCF. The decrease in the O$_3$ burden (5 %) under SSP3-7.0-lowCH4 is slightly less than that under SSP3-7.0-lowNTCF (7 %), and shows that reductions in CH$_4$ concentrations effectively reduce the tropospheric O$_3$ burden despite high NTCF emissions under SSP3-7.0-lowCH4. O$_3$ production rates under SSP3-7.0-lowCH4 are slightly higher than in the present day partly due to higher hydroxyl radical (OH) concentrations that promote O$_3$ production. However, these higher O$_3$ production rates are offset by higher O$_3$ loss rates, and result in lower O$_3$ net production under SSP3-7.0 and SSP3-7.0-lowCH4. We find that the O$_3$ chemical lifetime decreases slightly by 0.4–1.6 days under future pathways partly due to decreased O$_3$ net production, and partly due to increased O$_3$ loss associated with higher temperature and humidity in a warmer climate (Young et al., 2018). Changes in O$_3$ dry deposition rates principally reflect changes in surface O$_3$ concentrations.





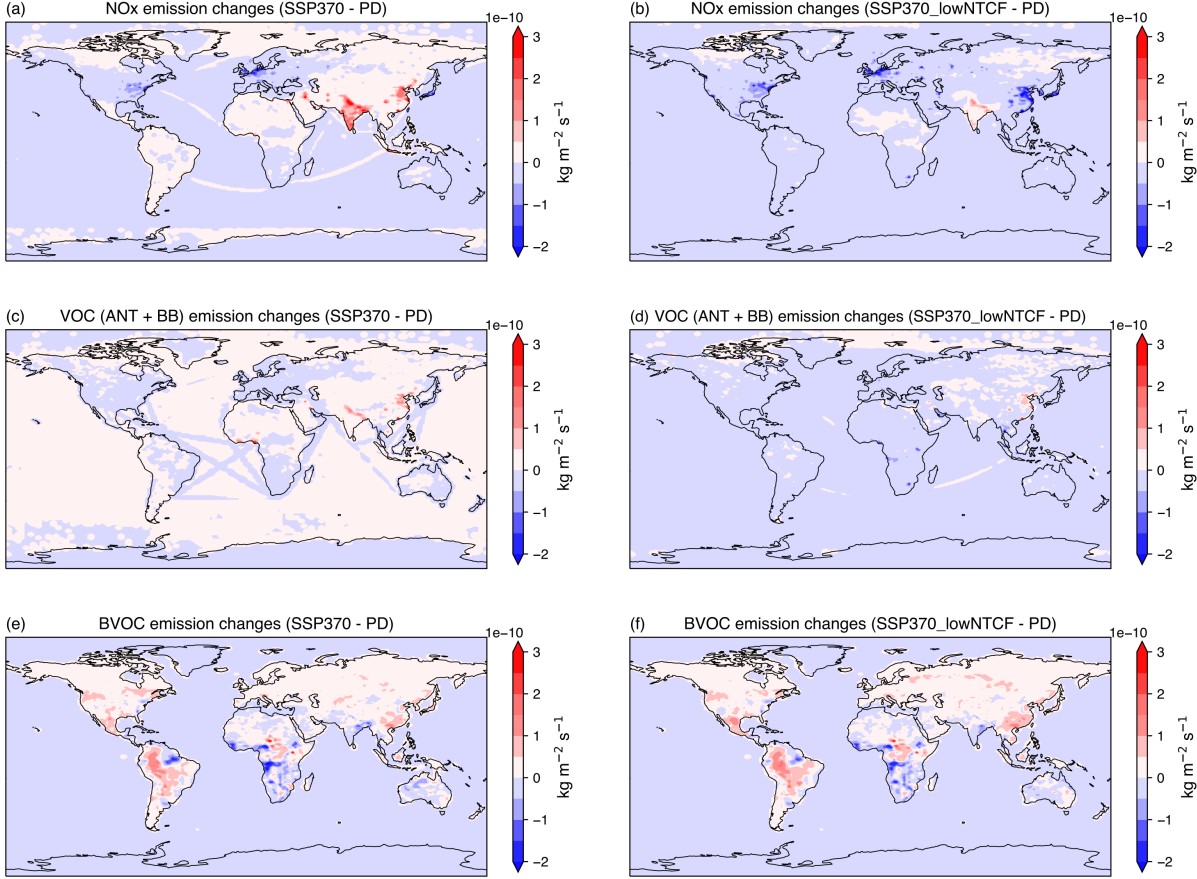

**Figure 3.** Differences in annual mean surface emissions of $NO_x$ **(a, b)**, anthropogenic and biomass burning VOCs **(c, d)** and total biogenic VOCs **(e, f)** between the present day (PD; 2004–2014) and SSP3-7.0 and SSP3-7.0-lowNTCF (2045–2055).

**Table 4.** Comparison of the tropospheric $O_3$ budget under present-day (2004–2014) and future conditions (2045–2055).

| | Present day | SSP3-7.0 | SSP3-7.0-lowNTCF | SSP3-7.0-lowCH4 |
|---|---|---|---|---|
| $O_3$ burden (Tg) | 376±3 | 393±5 | 351±7 | 357±6 |
| $O_3$ lifetime (days) | 22.5±0.2 | 20.9±0.1 | 22.1±0.1 | 21.1±0.2 |
| $O_3$ net production (Tg year$^{-1}$) | 996±40 | 934±38 | 638±60 | 869±32 |
| $O_3$ production (Tg year$^{-1}$) | 6080±66 | 6728±43 | 5536±108 | 6108±65 |
| $O_3$ loss (Tg year$^{-1}$) | 5084±59 | 5795±45 | 4898±78 | 5239±56 |
| $O_3$ dry deposition (Tg year$^{-1}$) | 936±9 | 961±9 | 810±19 | 856±14 |



### 4.3 Surface seasonal $O_3$ changes

Seasonal differences in simulated surface $O_3$ concentrations between the present day (2004–2014) and future pathways (2045–2055) are shown in Fig. 4a–f, along with a comparison between SSP3-7.0 and SSP3-7.0-lowCH4 (Fig. 4g, h). SSP3-7.0

represents less stringent emission control policies, and has slightly higher global mean $O_3$ concentrations (0.7–0.9 ppb; Fig. 4a, b) than the present day. In contrast, tightened emission controls under SSP3-7.0-lowNTCF reduce surface $O_3$ concentrations substantially across many parts of the world (3.3–5.2 ppb; Fig. 4c, d). The reduction in $CH_4$ concentrations from 1803 to 1364 ppb under SSP3-7.0-lowCH4 relative to the present day successfully reduces $O_3$ concentrations (2.7–3.5 ppb; Fig. 4e, f) for regions that show $O_3$ increases under SSP3-7.0. SSP3-7.0-lowCH4 with reduced $CH_4$ concentrations alone shows uniform $O_3$

decreases across the globe (3.4–4.4 ppb; Fig. 4g, h) compared with SSP3-7.0, where mean $CH_4$ concentrations are 2472 ppb, and this offsets high $O_3$ levels in regions with high NTCF emissions. This demonstrates the importance of $CH_4$ in governing surface $O_3$ concentrations, and the need to account for $CH_4$ in mitigating $O_3$ pollution in future (Fiore et al., 2008; Allen et al., 2021). We highlight that $O_3$ changes vary by season. From Fig. 4a, c we can see that $O_3$ concentrations generally increase in winter in continental areas such as North America and Europe under SSP3-7.0 and SSP3-7.0-lowNTCF. These regions have

large reductions in NTCF emissions under future pathways, and thus there is less $O_3$ titration due to lower $NO_x$ emissions in these regions. Since $NO_x$ emissions decrease in East Asia under SSP3-7.0-lowNTCF, we also see $O_3$ increases in winter in this region. This highlights that $NO_x$ emission reductions are not beneficial for reducing surface $O_3$ concentrations in winter. Conversely, $NO_x$ emission increases under SSP3-7.0 lead to $O_3$ decreases in winter in South and East Asia for the same reason. The situation is different in summer, with $O_3$ decreases in North America and Europe, but $O_3$ increases in South and East

Asia under SSP3-7.0 (Fig. 4b). This demonstrates a shift in $O_3$ sensitivity from VOC limitation in winter to $NO_x$ limitation in summer. $O_3$ changes in summer and winter are generally consistent in South America and Africa, and reflect $NO_x$ limitation in these regions throughout the year. For regions that are projected to have lower $NO_x$ emissions in SSP3-7.0-lowNTCF, such as eastern China, we note that $O_3$ concentrations increase in both winter and summer. The industrial regions of China are in VOC limited regimes throughout the year, and thus decreased $NO_x$ emissions increase $O_3$ concentrations (Jin and Holloway,

2015; Wang et al., 2021; Liu et al., 2021). This suggests that reductions in both $NO_x$ and VOC emissions may be needed to reduce $O_3$ in these regions.

We also examine daytime and nighttime $O_3$ changes for different regions under future pathways because daytime $O_3$ concentrations are typically more relevant for human health. For daytime $O_3$ concentrations we consider the maximum daily average 8 h $O_3$ concentration (MDA8), an important metric used to evaluate $O_3$ impacts on human health. Nighttime $O_3$ concentrations

are correspondingly given by the minimum daily average 8 h $O_3$ concentration (MIN8). We show a comparison of the changes in these metrics in Fig. 5.

In general, differences in global mean $O_3$ changes are relatively small in both daytime and nighttime under all pathways. However, in South and East Asia with increased $NO_x$ emissions under SSP3-7.0 and SSP3-7.0-lowCH4, $O_3$ concentrations tend to increase in daytime but show smaller increases or reduction at night. This demonstrates the impact of $O_3$ titration by NO at nighttime in high-$NO_x$ environments. In North America and Europe that have lower $NO_x$ emissions in the future,



**Figure 4.** Differences in seasonal mean surface $O_3$ concentrations between the present day (PD; 2004–2014) and the future SSP3-7.0 **(a, b)**, SSP3-7.0-lowNTCF **(c, d)** and SSP3-7.0-lowCH4 **(e, f)** (2045–2055) in winter (DJF) and summer (JJA). Differences between SSP3-7.0 and SSP3-7.0-lowCH4 are shown **(g, f)** to isolate the impacts of reduced $CH_4$ concentrations. Absolute global mean $O_3$ changes (ppb) are shown at the right top of each panel.

daytime $O_3$ concentrations decrease greatly in summer, but daytime and nighttime changes are similar in winter, demonstrating the large influence of $NO_x$ emissions on summer daytime $O_3$ concentrations. The substantial differences between daytime and nighttime $O_3$ changes suggest that the underlying impacts of $O_3$ changes on human health are likely to be larger than those





estimated using seasonal mean $O_3$ concentrations, and this is particularly important for high-emission areas with increased

$NO_x$ emissions in the future.



**Figure 5.** Seasonal mean maximum daily average 8 h (MDA8) and minimum daily average 8 h (MIN8) surface $O_3$ changes for North America, Europe, South Asia, East Asia and the globe from the present day (PD; 2004–2014) to SSP3-7.0 **(a)**, SSP3-7.0-lowNTCF **(b)** and SSP3-7.0-lowCH4 **(c)** (2045–2055). DJF and JJA situations are shown with blue and red bars, respectively.

# 5 $O_3$ sensitivity in the present day and the future

Non-linearity in chemical $O_3$ formation can result in differences in the effectiveness of emission control strategies regionally, and may aggravate $O_3$ pollution issues. We therefore investigate $O_3$ sensitivity for the present day and the future, as it is

10.5194/acp-2021-689
Atmospheric Chemistry and Physics



important to understand how $O_3$ concentrations will respond to changing emissions. Ratios of $NO_x$ and VOC concentrations

provide a useful indicator of regional $O_3$ sensitivity regimes. Here we quantify the critical $NO_x$/VOC ratio that distinguishes VOC-limited and $NO_x$-limited regimes by examining monthly mean surface $O_3$ concentrations and net chemical production rates as a function of monthly mean $NO_x$ and VOC concentrations, see Fig. 6. Monthly mean data from all months and all scenarios are used to plot the isopleths. Approximate thresholds of monthly mean $NO_x$/VOC ratios for $O_3$ sensitivity are shown in Fig. 6, ranging from 0.6 to 1 with a central value of 0.8. We hence apply a threshold value of 0.8 to distinguish $O_3$

sensitivity regimes hereafter. Areas above the threshold represent VOC-limited regimes in which increased $NO_x$ emissions reduce $O_3$ concentrations and $O_3$ production rates, and areas below the threshold represent $NO_x$-limited regimes. We find the highest $O_3$ concentrations and $O_3$ production close to the threshold line, demonstrating that the threshold values are consistent and that the approach is robust.

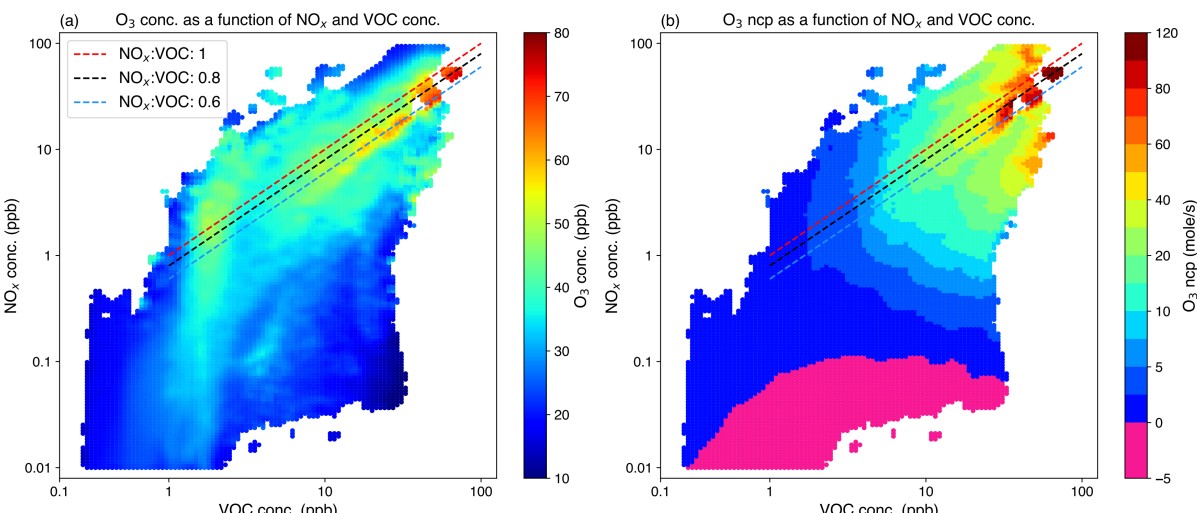

**Figure 6.** Surface $O_3$ concentrations **(a)** and $O_3$ net chemical production rates **(b)** as a function of monthly mean $NO_x$ and VOC concentrations. Monthly mean data for all months and all scenarios are used. The straight lines show the approximate thresholds of $NO_x$/VOC to distinguish VOC-limited (above the line) and $NO_x$-limited (below the line) regimes.

We further investigate $O_3$ sensitivity for different regions. The regions considered here are those defined for the Task Force

on Hemispheric Transport of Air Pollutants Phase 2 (TF HTAP2; Janssens-Maenhout et al., 2015), see Fig. 7a; the regions dominating each part of the $NO_x$-VOC concentration space are shown in Fig. 7b, c. We identify the dominant region based on the number of model grid cells with the corresponding $NO_x$ and VOC concentrations. This approach reveals differences in regional $O_3$ sensitivity. We also show the shift in $O_3$ sensitivity in different regions between the present day (Fig. 7b) and the future (SSP3-7.0-lowNTCF; Fig. 7c) to demonstrate the impacts of decreased NTCF emissions on the evolution of $O_3$

sensitivity on a regional basis.





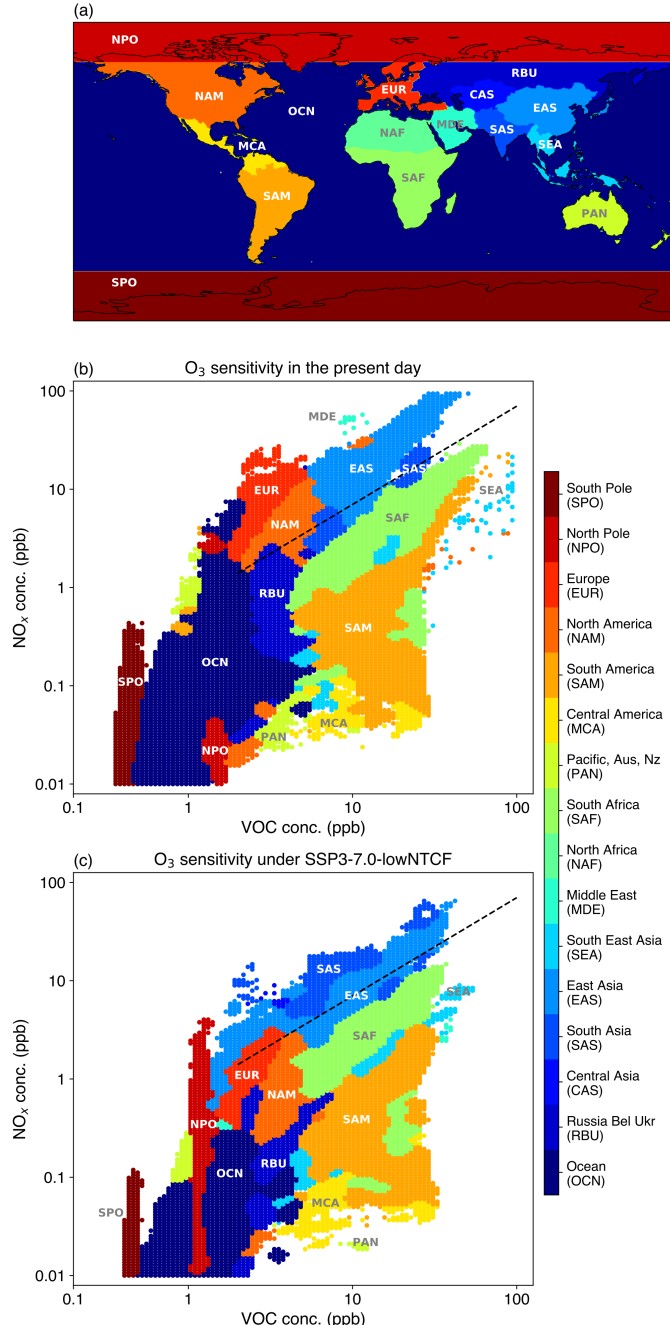

**Figure 7.** Geographical regions defined in the Task Force on Hemispheric Transport of Air Pollutants Phase 2 (TF HTAP2) **(a)**. Regions dominating each part of $NO_x$-VOC concentration space in the present day **(b)** and under SSP3-7.0-lowNTCF **(c)**. The straight line shows the $NO_x$/VOC threshold of 0.8. Monthly mean data for all months under each pathway are used to plot **(b)** and **(c)**.



Figure 7b clearly shows that low-$NO_x$ and low-VOC environments are most common in oceanic and polar regions where surface $O_3$ levels are typically low and where $O_3$ production is $NO_x$-limited. In contrast, Europe, North America and East Asia dominate the high-$NO_x$ and high-VOC environments in the present day where $O_3$ levels are high and $O_3$ production is VOC-limited. Europe and North America have similar VOC concentrations but $NO_x$ concentrations are generally higher in

Europe than in North America, which results in Europe lying further above the $NO_x$/VOC threshold. This demonstrates that stricter controls on $NO_x$ emissions are required for Europe to shift from VOC-limited to $NO_x$-limited regimes than for North America. East Asia dominates VOC-limited regimes due to much higher $NO_x$ and VOC concentrations than other regions. Parts of South Asia and Middle East are also VOC-limited. Major biogenic emission source regions such as South America, South Africa and South East Asia have high VOC concentrations but moderate levels of $NO_x$, and the chemical environment

is therefore $NO_x$-limited.

The impacts of reductions in NTCF emissions are shown in Fig. 7c. We find that Europe and North America are no longer the most dominant VOC-limited regimes due to decreased $NO_x$ concentrations. East Asia is still a dominant VOC-limited region under SSP3-7.0-lowNTCF, but reduced $NO_x$ emissions shift parts of East Asia into $NO_x$-limitation. South Asia becomes the main VOC-limited region with relatively high $NO_x$ concentrations. South America, South Africa and South East Asia are still

$NO_x$-limited because there are no large $NO_x$ increases in these regions.

$O_3$ sensitivity in major present-day VOC-limited regions under different scenarios are shown in Fig. 8. Reductions in $NO_x$ emissions are important and effective in transforming VOC-limitation to $NO_x$-limitation, reflected in large $NO_x$-limited areas in Europe and North America under all scenarios. In contrast, most parts of East Asia are VOC-limited under SSP3-7.0 and SSP3-7.0-lowCH4 due to increased $NO_x$ emissions. Since changes in VOC concentrations are relatively small for all scenarios,

they have little substantial influence on $O_3$ sensitivity. Under future climate, increased biogenic VOC emissions in Europe and North America offset decreased anthropogenic and biomass VOC emissions, and are hence beneficial to maintain a $NO_x$-limited environment. We note that South Asia is the only region that is substantially VOC-limited in the future due to increased $NO_x$ emissions. Reductions in $CH_4$ concentrations have relatively little impact on $O_3$ sensitivity, although they greatly reduce surface $O_3$ concentrations (Sect. 4.3).



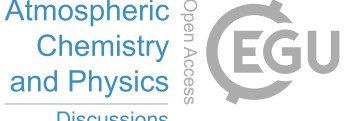

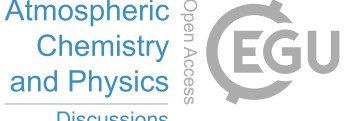

**Figure 8.** $O_3$ sensitivity across Europe, North America, East Asia and South Asia in the present day **(a)** and under SSP3-7.0 **(b)**, SSP3-7.0-lowNTCF **(c)** and SSP3-7.0-lowCH4 **(d)**. The $NO_x$/VOC threshold of 0.8 is shown. Horizontal and vertical lines indicate regional mean $NO_x$ and VOC concentrations in the present day.





## 6  Spatial distributions of O$_3$ sensitivity

Global spatial distributions of annual O$_3$ sensitivity in the present day and the future are shown in Fig. 9. VOC-limited regimes are represented by high NO$_x$/VOC ratios, reflecting relatively high NO$_x$ or low VOC concentrations. In the present day, high NO$_x$ emissions contribute to VOC-limitation in large areas of North America, Western and Central Europe and East Asia. While North America and Europe have lower NO$_x$ concentrations than East Asia, lower VOC concentrations still lead to VOC-limitation. Only south-west parts of India are VOC-limited. In the future, O$_3$ production in more areas of North America and Europe becomes NO$_x$-limited. However, VOC limited regimes in East Asia, particularly China, are persistent due to projected increases in NO$_x$ emissions until 2055 under SSP3-7.0 and SSP3-7.0-lowCH4. We find that reductions in CH$_4$ concentrations have relatively little influence on O$_3$ sensitivity over continental regions (Fig. 9b vs 9d). We note that SSP3-7.0-lowNTCF shows the smallest VOC-limited areas across the globe. East Asia is still partly VOC-limited under this scenario, particularly in northern China (Fig. 9c), which indicates that further reductions in NO$_x$ emissions are required in addition to those expected in this region in the future (Fig. 3b).

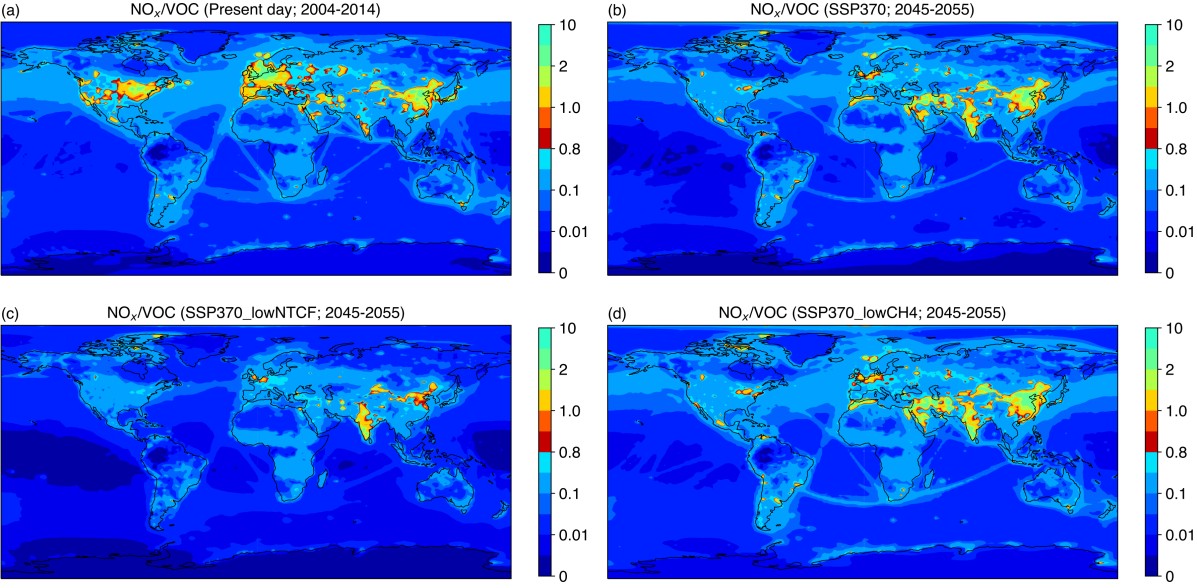

**Figure 9.** Spatial distributions of annual mean NO$_x$/VOC in the present day (2004–2014; **a**) and under SSP3-7.0 (**b**), SSP3-7.0-lowNTCF (**c**) and SSP3-7.0-lowCH4 (**d**) (2045–2055). A NO$_x$/VOC threshold ratio of 0.8 is used here to distinguish O$_3$ sensitivity regimes.

We contrast regional O$_3$ sensitivities for winter and summer seasons in Fig 10, as we note that O$_3$ responses in different seasons can be substantially different in high-emission regions. This suggests that static emission control strategies throughout the year may not be the best way to lower annual-mean O$_3$ pollution, and adjustments may be needed according to seasonal O$_3$ sensitivities. More extensive VOC-limited areas are found in winter than in summer under both present day and future conditions, but these account for less than 7 % of the total area of the world in the present day (Table 5). Over 50 % of North





America, Europe and East Asia are VOC-limited in winter in the present day. This explains why North America and Europe show $O_3$ increases in winter (Fig. 4) despite reduced $NO_x$ emissions. However, there are fewer VOC-limited regions across the globe under all future pathways (Fig. 10). About 1 % of North America is VOC-limited in summer, and less than 7 % of

Europe. In contrast, over 48 % (winter) and 39 % (summer) of South Asia is VOC-limited in future. Slightly more areas are VOC-limited under SSP3-7.0-lowCH4 than under SSP3-7.0 (Fig 10c, d). Overall, reductions in $NO_x$ emissions are important to reduce $O_3$ production in high-emission regions and shift VOC-limited areas to $NO_x$-limitation but this may lead to higher $O_3$ concentrations in winter without further emission controls on VOC and $CH_4$.

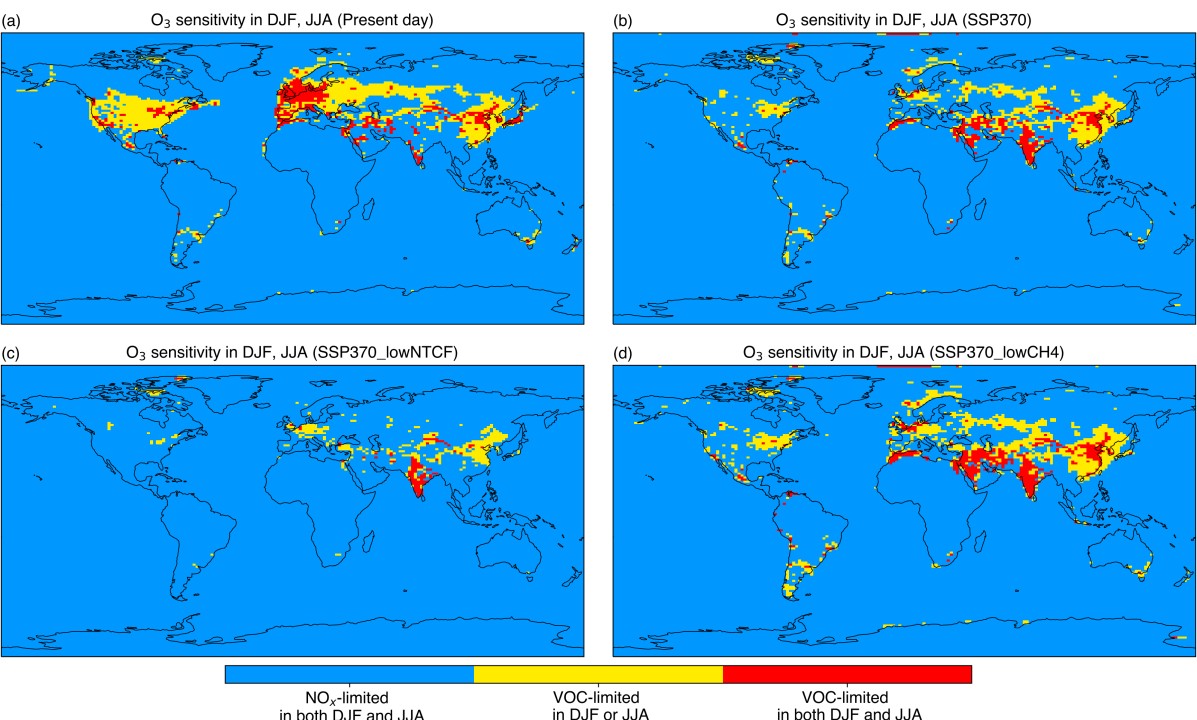

**Figure 10.** Seasonal differences in $O_3$ sensitivity regimes for the present day (2004–2014; **a**) and under SSP3-7.0 (**b**), SSP3-7.0-lowNTCF (**c**), SSP3-7.0-lowCH4 (**d**) (2045–2055). A $NO_x$/VOC threshold ratio of 0.8 is used here to distinguish $O_3$ sensitivity regimes.





**Table 5.** The percentage of VOC-limited areas (%) in different regions under different pathways in winter (DJF) and in summer (JJA).

|  | Present day | | SSP3-7.0 | | SSP3-7.0-lowNTCF | | SSP3-7.0-lowCH4 | |
|---|---|---|---|---|---|---|---|---|
|  | DJF | JJA | DJF | JJA | DJF | JJA | DJF | JJA |
| N. America | 50.4 | 6.4 | 12.5 | 0.4 | 2.4 | 0 | 16.9 | 1.0 |
| Europe | 79.8 | 37.4 | 27.0 | 2.6 | 27.5 | 0.9 | 41.6 | 6.1 |
| S. Asia | 15.1 | 15.3 | 54.8 | 49.6 | 48.1 | 39.5 | 60.4 | 56.7 |
| E. Asia | 60.3 | 18.3 | 63.3 | 15.8 | 37.7 | 3.5 | 68.7 | 19.3 |
| Globe | 6.9 | 2.7 | 4.8 | 2.0 | 2.0 | 0.6 | 5.9 | 3.0 |

## 7 Conclusions

We use a global chemistry-climate model, UKESM1, to assess the impacts of changing near-term climate forcer (NTCF) emissions and $CH_4$ concentrations in the context of climate change on tropospheric $O_3$ in the present day (2004–2014) and the near future (2045–2055). CMIP6 future scenarios representing 'regional rivalry' development pathways (SSP3-7.0, SSP3-7.0-lowNTCF and SSP3-7.0-lowCH4) are used from the AerChemMIP project. We have examined $O_3$ changes from the present day to the future and investigated regional $O_3$ sensitivities to explain contrasting $O_3$ changes in different seasons.

An extended chemistry scheme incorporating more reactive VOC species is used to permit representation of more active photochemical environments for $O_3$ production. This shows higher surface $O_3$ concentrations in high-emission regions and a 5 % higher tropospheric $O_3$ burden. While simulated surface $O_3$ concentrations are biased low in winter and high in summer, these systematic model biases are similar to those using the original chemistry scheme as well as a more comprehensive chemistry scheme. This indicates that other factors in the model are likely to be responsible for the biases but the extended

chemistry scheme permits representation of a more appropriate chemical environment for $O_3$ production in high-emission areas.

From the present day to the future, the tropospheric $O_3$ burden increases by 4 % under SSP3-7.0 and decreases by 7 % and 5 % under SSP3-7.0-lowNTCF and SSP3-7.0-lowCH4. The tropospheric $O_3$ chemical lifetime remains similar (21–22 days) under all scenarios, and this is similar to previous estimates. Seasonal global mean surface $O_3$ concentrations increase

by 0.7–0.9 ppb under SSP3-7.0, and decrease by 3.3–5.2 ppb under SSP3-7.0-lowNTCF and by 2.7–3.5 ppb under SSP3-7.0-lowCH4. We find that reductions in NTCF emissions are effective in reducing surface $O_3$ concentrations, and reductions in $CH_4$ concentrations are also important. We also find that both the magnitude and direction of seasonal, daytime and nighttime $O_3$ changes relative to the present day can vary greatly across different regions especially South and East Asia.

$O_3$ sensitivity is quantified using monthly mean $NO_x$/VOC concentration ratios to give a broad assessment of regional

$O_3$ sensitivity. The estimated monthly mean $NO_x$/VOC thresholds range from 0.6 to 1.0, and 0.8 is used to distinguish $O_3$ sensitivity regimes. Most VOC limited regimes occur in high-emission regions across the northern hemisphere, such as North



America, Europe, the Middle East, South Asia and East Asia. More areas in North America and Europe become $NO_x$-limited under all future pathways due to the projected decrease in $NO_x$ emissions. There are more VOC-limited areas in East Asia under SSP3-7.0 and SSP3-7.0-lowCH4 due to the projected increase in $NO_x$ emissions, although there are fewer VOC-limited areas in East Asia under SSP3-7.0-lowNTCF. South Asia becomes the dominant region for VOC-limited $O_3$ production in the future. Projections of regional $O_3$ sensitivity demonstrate that reductions in $NO_x$ emissions are the most important factor to shift VOC-limited regimes to $NO_x$-limitation.

We highlight that $O_3$ sensitivity varies by season. There are more VOC-limited regimes in winter (7 %) than in summer (3 %) across the globe. Reductions in $NO_x$ emissions increase surface $O_3$ concentrations in high-emission areas particularly in winter, and reductions in VOC emissions should be targeted. In the future, reductions in $NO_x$ and VOC emissions should both be effective in mitigating $O_3$ pollution in most areas of North America and Europe in summer because there are only 1 % and 7 % VOC-limited areas in these two regions. However, further reductions in $NO_x$ emissions are needed for parts of East Asia and South Asia to convert most VOC-limited areas to $NO_x$-limited. While anthropogenic and biomass emissions may be controlled in the future, more biogenic emissions under a warmer climate would hinder the impacts of VOCs on $O_3$ mitigation. Reductions in $CH_4$ concentrations are also important to reduce surface $O_3$ pollution. $NO_x$ decreases are important to reduce surface $O_3$ concentrations from a globe perspective but will lead to increased $O_3$ concentrations in some regions, and hence emission controls on VOC and $CH_4$ are necessary to mitigate regional $O_3$ pollution during the transition from VOC- to $NO_x$-limitation.

*Data availability.* The data generated in this study are available upon request.

*Author contributions.* ZL, RD, OW, FO'C, ST designed the study. ZL set up the model, conducted model simulations and performed the analysis. ZL, RD and OW prepared the paper. All co-authors contributed to reviewing and editing the paper.

*Competing interests.* The authors declare that they have no conflict of interest.

*Acknowledgements.* Zhenze Liu thanks the University of Edinburgh China Scholarship Council. Oliver Wild and Ruth M. Doherty thank the Natural Environment Research Council (NERC) for funding under grants NE/N006925/1, NE/N006976/1 and NE/N006941/1. Steven T. Turnock thanks the UK-China Research & Innovation Partnership Fund through the Met Office Climate Science for Service Partnership (CSSP) China as part of the Newton Fund. This work made use of computation resources on the Met Office and NERC joint supercomputer system (MONSooN) in the UK.





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
