# Peer review of "Tropospheric ozone changes and ozone sensitivity from present-day to future under shared socio-economic pathways"

_Atmospheric Chemistry and Physics, 2021_

## Author Comment (AC1)

Dear editor and all reviewers:

We thank the editor and all reviewers for their contribution to the improvement of the ACP manuscript. Responses to reviewers on "Tropospheric ozone changes and ozone sensitivity from present-day to future under shared socio-economic pathways" by Zhenze Liu et al. are given below. For clarity, the reviewer comments are given in bold, followed by our responses. The modified text in our revised manuscript is given in quotes, italics and blue.

**Response to reviewer 1:**

1. **This manuscript explores the sensitivity of ozone production in a future climate across three possible emissions scenarios. The paper is extremely well written, the figures and tables are clear and self-explanatory and the conclusions are sound and supported by the evidence. I have a few minor comments, mainly focused on the need to cite previous work, and to present the findings in the context of the new conclusions by IPCC AR6.**

We thank the reviewer for your positive comments here, and address specific concerns below.

2. **The paper would benefit from some discussion that places these results in the broad context of the recent findings of IPCC AR6. I realize that AR6 was not published when this paper was submitted, so AR6 could not be referenced. But now that AR6 is publicly available, a comparison is warranted, especially in terms of Chapter 6, Section 6.5.1, "Effect of climate change on ozone". The broad message from this new analysis is that ozone in the mid-21st century will be lower across the USA than it is today, under all three scenarios. Presumably, the primary cause is the decrease in regional US emissions of NOx. However, part of the explanation could also be due to a shorter ozone lifetime in a warmer, more humid future. How much of this decrease is due to emissions changes and how much is due to climate change? Along these same lines, what is the impact of future heat waves, which are expected to be more intense in the future? Stronger heat waves will lead to episodic surface ozone pollution events. Will these future pollution episodes be more intense than present day events? Is the average decline in ozone masking a few extreme high ozone events in the future?**

We thank the reviewer for highlighting results from the latest IPCC AR6 report. Surface $O_3$ will be influenced by many factors associated with climate change, but we note that the influences can vary at regional and continental scales (Doherty et al., 2013; Doherty et al., 2017). We find that reductions in $NO_x$ emissions largely explain surface $O_3$ decreases in high-emission regions in summer in the future, as the reviewer notes, but also expect changes in surface $O_3$ concentrations due to higher average temperatures that impact $O_3$ production and destruction rates, and temperature-sensitive precursor emissions, and to more frequent heat waves and stagnation under a warmer climate. We have examined the distributions of summertime surface hourly mean $O_3$ concentrations in different future scenarios, as shown in Fig. 1. Increases in extremely high $O_3$ concentrations under SSP370 (Fig. 1a) are principally due to increased $O_3$ precursor emissions. However, there are no significant

increases in the frequency of extremely high O₃ concentrations under SSP3-7.0-lowNTCF and SSP3-7.0-lowCH4 (Fig. 1b, c), which demonstrates that reductions in O₃ precursor emissions offset high surface O₃ levels associated with heat waves and stagnation in the future. We have added some discussion about climate change in the paper:

[Figure]

**Figure 1.** Frequency distributions of global surface hourly mean O₃ mixing ratios in summer (July) in the present day (2004-2014) and future scenarios (2045-2055).

Page 10, line 205:

*"Changes in O₃ dry deposition rates principally reflect changes in surface O₃ concentrations, although high temperatures under a warmer climate may reduce O₃ deposition rates due to vegetation stress (Lin et al., 2020)."*

Page 12, line 232:

*"Surface O₃ concentrations are also influenced by climate change, reflecting changing natural emissions, O₃ production and destruction rates and O₃ deposition rates (Doherty et al., 2013; Doherty et al., 2017). Global annual mean surface O₃ mixing ratios decrease by 1 ppb with a 1.5 °C temperature rise, but show little change in continental areas (Naik et al., 2021). This is principally due to increased humidity and greater O₃ destruction in oceanic areas, but in continental areas these effects may be offset by O₃ increases due to higher soil NOₓ (Romer et al., 2018) and BVOC emissions, and by decreased O₃ deposition rates (Lin et al., 2020). O₃ concentrations can also be impacted by more frequent and intense heat waves under a warmer climate (Schnell and Prather, 2017; Ma et al., 2019). We find that the resulting changes in surface O₃ concentrations in continental regions due to climate change are relatively small, and reduction in anthropogenic emissions is the dominant factor governing surface O₃ concentrations in the near future."*

3. **In the Introduction (lines 36-41) several of the papers cited in terms of describing recent ozone trends are out of date. For example, the data analysis in Lefohn et al. (2008) stops in 2005 so the paper does not report the strong decreases of ozone in the eastern USA that occurred after 2004. The paper by Akimoto et al. (2003) does not even report observations from the 21st Century. Ohara et al., 2007 is also out of date. Current papers are:**

**Simon, H, Reff, A, Wells, B, Xing, J and Frank, N (2015), Ozone Trends Across the United**

States over a Period of Decreasing NOx and VOC Emissions. Environ. Sci. Technol 49: 186–195. DOI: https://doi.org/10.1021/es504514z

Strode, S. A., J. M. Rodriguez, J. A. Logan, O. R. Cooper, J. C. Witte, L. N. Lamsal, M. Damon, B. Van Aartsen, S. D. Steenrod, and S. E. Strahan (2015), Trends and variability in surface ozone over the United States, J. Geophys. Res. Atmos., 120, 9020–9042, doi:10.1002/2014JD022784

Lu, X., Zhang, L., Wang, X., Gao, M., Li, K., Zhang, Y., Yue, X. and Zhang, Y., 2020. Rapid increases in warm-season surface ozone and resulting health impact in China since 2013. Environmental Science & Technology Letters, 7(4), pp.240-247.

Chang, K-L, I. Petropavlovskikh, O. R. Cooper, M. G. Schultz and T. Wang (2017), Regional trend analysis of surface ozone observations from monitoring networks in eastern North America, Europe and East Asia, Elem Sci Anth., 5:50, DOI:http://doi.org/10.1525/elementa.243

Tarasick, D. W., I. E. Galbally, O. R. Cooper, M. G. Schultz, G. Ancellet, T. Leblanc, T. J. Wallington, J. Ziemke, X. Liu, M. Steinbacher, J. Staehelin, C. Vigouroux, J. W. Hannigan, O. García, G. Foret, P. Zanis, E. Weatherhead, I. Petropavlovskikh, H. Worden, M. Osman, J. Liu, K.-L. Chang, A. Gaudel, M. Lin, M. Granados-Muñoz, A. M. Thompson, S. J. Oltmans, J. Cuesta, G. Dufour, V. Thouret, B. Hassler, T. Trickl and J. L. Neu (2019), Tropospheric Ozone Assessment Report: Tropospheric ozone from 1877 to 2016, observed levels, trends and uncertainties. Elem Sci Anth, 7(1), DOI: http://doi.org/10.1525/elementa.376

Gaudel, A., O. R. Cooper, K.-L. Chang, I. Bourgeois, J. R. Ziemke, S. A. Strode, L. D. Oman, P. Sellitto, P. Nédélec, R. Blot, V. Thouret, C. Granier (2020), Aircraft observations since the 1990s reveal increases of tropospheric ozone at multiple locations across the Northern Hemisphere. Sci. Adv. 6, eaba8272, DOI: 10.1126/sciadv.aba8272

We thank the reviewer for these references. Some old citations have been removed and replaced with more recent references:

Page 2, line 38:
*"In recent decades, there has been a decrease in surface $O_3$ concentrations in North America and Europe due to emission controls (Simon et al., 2015; Colette et al., 2016; Tarasick et al., 2019). In contrast, increases in surface $O_3$ levels are observed in South Asia and East Asia due to industrialization, urbanization and social development (Hakim et al., 2019; Lu et al., 2020)."*

4. **Table 2 indicates that methane will be about 1364 ppb in 2050 under the SSP3-7.0-lowCH4, which is far lower than the present-day value of about 1890 ppb (https://gml.noaa.gov/ccgg/trends_ch4/). This would require major reductions in CH4. To**

**provide context for the reader, can you let us know when methane was last at such a low level in the atmosphere? I'm guessing that it would be sometime around the 1960s based on Figures 2.4 and 2.5 of IPCC AR6. According to Figure 2.5 of IPCC AR6, methane was approximately 1500 ppb in the 1970s, based on in situ observations. According to Figure 2.4 of IPCC AR6, methane was about 1000 ppb in the early 20th century (perhaps around 1940?).**

The last time that surface $CH_4$ concentrations were ~1360 ppb was about 1970 (Prather et al., 2014), which is visualised at https://www.methanelevels.org/. We have now provided this relevant context in the paper:

Page 4, line 110:
*"SSP3-7.0-lowCH4 follows SSP3-7.0 but assumes strong mitigation of $CH_4$ emissions in the future, with 24 % decreases in surface $CH_4$ mixing ratios from 1802 ppb to 1364 ppb. The last time that historical surface $CH_4$ mixing ratios were this low was more than 50 years ago, in the late 1960s (Prather et al., 2014)."*

5. **It would help to briefly mention the significance of the term "regional rivalry" when describing the future scenarios. This term is mentioned twice in the paper, so it must have some importance, but I really have no idea what it means.**

"Regional rivalry" is just a descriptive label for the whole set of SSP3 pathways. It represents competition and conflicts between countries for energy and food supplies in the future, leading to strong environmental degradation in some regions and a warmer climate. To avoid confusion, we have removed this term from the paper.

6. **When using the TOAR data products, the following data link should be cited, in addition to the peer-reviewed publication by Schultz et al. (2017): Schultz, M. G, et al. (2017): Tropospheric Ozone Assessment Report, links to Global surface ozone datasets. PANGAEA, https://doi.org/10.1594/PANGAEA.876108**

We have now cited this data publication.

Page 8, line 163:
*"We now evaluate surface $O_3$ concentrations simulated with the Ext_StratTrop chemistry scheme against gridded monthly mean rural observations from the TOAR dataset over the 2004–2014 period (Schultz et al., 2017a, b)."*

Page 9, line 179:
*"Observations from the Tropospheric Ozone Assessment Report (TOAR) dataset (Schultz et al., 2017b) are used for comparison."*

**7. Line 321 globe should be global**

Corrected.

**8. Figure 6 and elsewhere**
   **When reporting trace gas values in units of ppb, one cannot use the term concentration, which is mass per volume. The expression mixing ratio must be used.**

We have corrected this. We use both 'concentration' and 'mixing ratio' in the paper, but replace all 'concentration' with 'mixing ratio' when relating to ppb.

**Response to reviewer 2:**

1. **Liu et al. examine the changes in tropospheric ozone between present day conditions and several future scenarios in model runs from UKESM1 as contributed to the AerChemMIP model intercomparison project. They show that the future tropospheric ozone burden and surface ozone mixing ratio are sensitive to the changes in the emissions of short-lived ozone precursors (NOx and NMVOC) and to changes in the assumed surface mixing ratio of methane. In the high-emissions SSP3-7.0 scenario the tropospheric ozone burden and the surface mixing ratio of ozone increases, while these decrease in variants of this scenario in which the emissions of ozone precursors or the methane mixing ratio decrease.**

   **Some high-emission regions show different trends to the trend in global surface ozone mixing ratio, which the authors relate to the modelled NOx-sensitivity of the ozone production regime in those areas, for example declining NOx emissions are associated with ozone surface mixing ratio increases in winter in all future scenarios in Europe and North America, but decreases in summer, consistent with NOx-saturated (or VOC-limited) conditions in these regions in present-day winter and NOx-limited conditions in present-day summer.**

   **So far, these results are not particularly novel. Where Liu et al. attempt to bring some novelty to the analysis is with a determination of the modelled local ozone production regime in individual model grid cells based on the modelled ratio of NOx and VOC mixing ratios (the authors call this "concentration", but they have clearly used molar mixing ratios in their calculations). Such an analysis is potentially interesting, as it would enable determination of the ozone production regime from a single model run rather than from a comparison of two runs. The authors do indeed do exactly this, and their results do make sense (eg. areas with high NOx emissions are generally NOx-saturated, and these can change to $NO_X$-limited as the NOx emissions are reduced). Unfortunately, the authors do not give enough information about their method to enable a proper understanding of it. This is itself unfortunate, since the novelty of the paper depends strongly on this analysis.**

We thank the reviewer for the comments and the recognition of the novelty in this study. The novelty arises from the analysis of $O_3$ sensitivity based on indicators over a global scale, and it is the first time that this analysis is applied with a global chemistry-climate model as far as we are aware. Regarding the application of $NO_x$/VOC ratios as an indicator of $O_3$ chemical regime, we have discussed the background to the $O_3$ sensitivity indicator used in the introduction but have now added further text to this paragraph for clarity as given below. In addition, we have added a new subsection 2.3 to the methods to describe the $O_3$ sensitivity indicators used. We had outlined the usage of $NO_x$/VOC ratios at the beginning of section 5 lines 258-261:

"Ratios of $NO_x$ and VOC concentrations provide a useful indicator of regional $O_3$ sensitivity regimes. Here we quantify the critical $NO_x$ /VOC ratio that distinguishes VOC-limited and $NO_x$-limited regimes by examining monthly mean surface $O_3$ concentrations and net chemical production rates as a function of monthly mean $NO_x$ and VOC concentrations."

We address the specific concerns about methods and novelty in detail below.

**2. Firstly, for such an important piece of analysis, the authors give no mention of any previous work that has performed it. This reviewer is not aware of any previous published attempt to quantify the ozone production regime in global model grid cells. If the authors are also similarly unaware of such work, they should say so! Alternatively, they should discuss their approach in the context of any previous work. And while there may not be a lot of literature on the simulated chemical regime in global model grid cells, there is certainly a mature literature on the general topic of ozone production regimes.**

We thank the reviewer for this comment. The traditional model-based approach to identifying $O_3$ sensitivity regimes is to investigate $O_3$ responses to changing $NO_x$ and VOC emissions. This is computationally demanding as it requires multiple model simulations. $O_3$ sensitivity indicators provide a simpler approach based on the ratios of chemical species related to $O_3$ production. However, most previous studies have focused on regional $O_3$ or on short time periods, and the critical values of $NO_x$/VOC ratios to distinguish sensitivity regimes may differ in different locations and time periods. To address this, we generalise the approach by investigating $NO_x$/VOC ratios from global and long-term perspectives. We have now acknowledged previous work and discussed the advantage of the approach used in this study, and we highlighted the novelty of the analysis in the introductory section of the paper:

Page 2, line 45:
*"$O_3$ sensitivity is typically characterised by $NO_x$ - or VOC-limited regimes for $O_3$ production, and this determines the effectiveness of different emission control strategies. It is dependent on the relative abundance of $NO_x$ and VOC concentrations (Sillman, 1999), or of their oxidation products, nitric acid ($HNO_3$) and hydrogen peroxide ($H_2O_2$) (Kleinman, 1994; Sillman, 1995)."*

Page 2, line 54:
*"$O_3$ sensitivity indicators such as the ratios of $NO_x$/VOCs and $HNO_3$/$H_2O_2$ allow us to identify $O_3$ sensitivity regimes relatively easily. However, most studies focus on $O_3$ sensitivity in specific regions and for short time periods (Dunker et al., 2002; Sillman and West, 2009; Ye et al., 2016), leading to inconsistency in the critical indicator values that distinguish $O_3$ sensitivity regimes. To address this, we generalise the approach by quantifying $O_3$ sensitivity using a consistent indicator across the globe. This is the first time that the full range of surface chemical environments across the globe has been explored with a global chemistry-climate model, as far as we are aware. We quantify $O_3$ sensitivity based on the ratio of $NO_x$ and VOC concentrations, and investigate how regional $O_3$ sensitivity might change in the future."*

**3. The authors define the ozone production regime based on the relative abundance of NOx and VOC, citing Sillman (1995) as a source for this. But Sillman (1995) discusses this regime based on the ratio of nitric acid and peroxides, based on the products of the NOx dependent dominant loss pathway for radical species; when NOx is high, the dominant**

**loss pathway is NO2 + OH -> HNO3, while when NOx is low, the dominant loss pathway is HO2 + HO2 -> H2O2 + O2. The relative abundance of HNO3 and H2O2 from radical termination reactions has in fact been used already in regional modelling studies as an indicator of the ozone production regime for the purpose of ozone source attribution (Dunker et al. 2002, Kwok et al. 2015). Did the authors consider a similar approach?**

**Kleinman (1994) showed that these regimes of radical loss are equivalent to the ozone production regimes and can also be characterised by the ratio of the NOx source to the radical source. The addition of VOC to a NOx-saturated system results in the production of carbonyl compounds (especially HCHO) which act as a source of radicals by their photolysis, hence the equivalence between "NOx-saturated" and "VOC-limited" chemical conditions.**

We choose a classic $NO_x$/VOC indicator in this study as it is easy to interpret and to apply. By using concentrations rather than emissions we avoid issues associated with the transport of species between model grid boxes. We considered the $HNO_3/H_2O_2$ concentration ratio in a previous study, and found that the $O_3$ sensitivity regimes determined with this indicator and with the $NO_x$/VOC concentration ratio we use here were similar (Liu et al., 2021). The $HNO_3/H_2O_2$ indicator is more sensitive to the chemistry scheme used, and there may be errors in the simulation of short-lived radicals (Whalley et al., 2021), which could lead to biases in identifying $O_3$ sensitivity regimes using this indicator. We also note that when gas and aerosol schemes are coupled the $HNO_3/H_2O_2$ indicator may be less reliable. We have discussed different indicators in the new subsection 2.3 as the reviewer suggests:

Page 4, line 117:
*"2.3 $O_3$ sensitivity indicators*
*A number of different indicators have been used to distinguish $O_3$ sensitivity regimes, and typical indicators are the ratios of $NO_x$/VOC concentrations or emissions and the ratio of $HNO_3/H_2O_2$ concentrations (Kleinman, 1994; Sillman, 1999). For the $NO_x$/VOC ratio, it is often more appropriate to use concentrations than emissions because this accounts for emissions, transport, chemical reactions and deposition. Indicators based on $HNO_3/H_2O_2$ concentration ratios also account for differences in photochemical conditions and VOC reactivity. In a previous study we found that $O_3$ sensitivity regimes diagnosed with $HNO_3/H_2O_2$ and $NO_x$/VOCs ratios were similar (Liu et al., 2021). However, the $HNO_3/H_2O_2$ indicator is more sensitive to uncertainties in chemical mechanism, and studies have shown that there are errors in the simulation of short-lived radicals in polluted areas (Whalley et al., 2021). The $HNO_3/H_2O_2$ ratio also does not account for gas-aerosol conversion as a termination route for $NO_x$, a mechanism that is included in many chemistry-climate models. We hence choose the ratio between $NO_x$ and VOC concentrations as a simple indicator of $O_3$ sensitivity indicator in this study.*

4. **So, the ozone production regime depends on more than just the local ratio of NOx and VOC. At any given ratio between NOx and VOC, the chemical system could be NOx saturated or NOx-limited depending on several other factors: the intensity of solar**

**radiation; the background abundance of ozone itself; and the OH reactivity of the VOC present. For example, the same ratio of NOx and VOC could lead to NOx-limited conditions in summer and NOx-saturated conditions in winter. Similarly, the same ratio of NOx and VOC could be NOx-limited if the VOC are highly reactive (eg. biogenic isoprene) but NOx saturated if the VOC are relatively unreactive (eg. most anthropogenic VOC). By defining the ozone production regime in terms of the relative abundance of NOx and VOC, the authors miss all this complexity.**

We agree that $O_3$ sensitivity is influenced by many factors but the impacts of these factors are also reflected in the concentrations of $O_3$ since these processes are simulated in the chemistry-climate model. Clearly there is sensitivity to insolation, temperature, humidity, VOC reactivity and other factors that vary both geographically and with season, and these factors partly explain the dominance of particular regions on the $O_3$ isopleths shown in Fig. 7. We have chosen to show both mean $O_3$ mixing ratios and $O_3$ net production rates on the isopleths in Fig. 6 to provide an overview of $O_3$ responses. We note that there is substantial variance in $O_3$ mixing ratios and $O_3$ net production rates across the $NO_x$-VOC space, as shown in the isopleths in Fig. 2 below, that reflects geographical, meteorological and seasonal influences. However, we average this to provide a broad overview of the underlying regimes, and we show the effect of seasonality by region in later sections. In Fig 6, we have aimed to characterise $O_3$ sensitivity in a general way that provides some insight from a global perspective, and we now state this aim clearly in the new subsection 2.3:

[Figure]

**Figure 2.** The ranges of surface **(a)** $O_3$ mixing ratios and **(b)** $O_3$ net chemical production rates as a function of monthly mean $NO_x$ and VOC mixing ratios. Monthly mean data for all months and for all scenarios are used.

Page 5, line 128:

*"We quantify the sensitivity of $O_3$ to $NO_x$ and VOC concentrations by examining monthly mean $O_3$ mixing ratios and $O_3$ net production in each UEKSM1 surface grid cell in each of the scenarios in turn. This provides a global overview of the dependence of $O_3$ and its production on $NO_x$ and VOC across different environments. It also allows us to determine a globally-averaged critical threshold value distinguishing $NO_x$-limited and VOC-limited regimes."*

5. **Another issue that the authors should address is the potential dependence of their ozone sensitivity metric on model resolution. The lifetime of NOx is relatively short, and the error of instantaneously diluting concentrated NOx emissions (characteristic of NOx-saturated regions) into relatively large grid cells has been well studied (eg. Wild et al., 2006).**

We thank the reviewer for pointing out this issue common to global models. Dilution of $NO_x$ and VOC emissions over coarse model grid cells generally leads to higher surface $O_3$ concentrations in polluted regions and may lead to overestimation of peak summertime $O_3$ in these regions. Since some VOC species have a substantially longer lifetime than $NO_x$, the actual $NO_x$/VOC ratios may be higher in polluted regions than we are able to resolve at coarse resolution. However, high-emission regions are typically already VOC-limited, and thus this bias does not greatly alter our identification of the different regimes. However, we acknowledge the importance of this effect and have added some relevant discussion in the subsection 2.3:

Page 5, line 132:
*"We note that dilution of short-lived $NO_x$ over coarse resolution model grid cells may lead to the underestimation of local $NO_x$ concentrations in high-emission regions. This results in underestimation of $NO_x$/VOC ratios in these conditions and the regimes may thus be more VOC-limited in reality than we are able to simulate in a global model."*

6. **This reviewer recognises that it may be challenging to diagnose the chemical regime in all grid cells of a global model from the relative strengths of the sources of NOx and radicals, and that perhaps the mixing ratios of HNO3 and H2O2 may not have been saved in the model output before this analysis. But the authors should at least discuss their approach and its limitations in the context of the previous literature on the topic. A good place for this discussion would be around the top of page 14, and an even better place for it would be collected in a new Subsection 2.3.**

We thank the reviewer for the suggestion. We have discussed our approach and its advantages in the new subsection 2.3 entitled "$O_3$ sensitivity indicators" as suggested. We have provided the text above in our response to Points 3-5.

7. **Along with this discussion, the authors should give more information about exactly what they did. Clearly the method has some use (as shown by the analysis in Sections 5 and 6), but for others to reproduce this work, more information is required. For example, which VOC are included in the quantification of the total VOC? Is it all VOC including intermediate oxidation products? Are there criteria for selecting which VOC to include? Is the OH reactivity of each individual VOC accounted for? Is methane included in the total VOC? If not, why not? What about CO? Which grid cells were used in the determination of the threshold ratio? Just the surface? Just the boundary layer? All the tropospheric grid cells?**

**I hope that the authors see that a lot more information is needed for the interested reader to understand what was done, how it relates to earlier work, how it could be reproduced, and what the potential limitations might be.**

We thank the reviewer for their advice. We have clarified our approach by adding more detail on the methods used in section 5:

Page 16, line 261:
*"Monthly mean $O_3$ mixing ratios and net production rates in the lowest model layer from all months and all scenarios are used to plot the figure. For $NO_x$ we use the sum of NO and $NO_2$ mixing ratios, and for total VOC we use the sum of the mixing ratios of primary emitted VOC species. CO and $CH_4$ are not included due to their relatively low reactivity. We classify $NO_x$ and VOC mixing ratios in each model grid cell into 150 bins on a logarithmic scale ranging from 0.01 ppb to 100 ppb, and calculate mean $O_3$ mixing ratios and mean $O_3$ net chemical production rates in each $NO_x$-VOC bin."*

8. **General comment**
   **The authors should avoid using the term "concentration" when they mean "mixing ratio" (multiple places throughout the text and figures).**

We have now corrected this. We use both 'concentration' and 'mixing ratio' in the paper, but replace all 'concentration' with 'mixing ratio' when relating to ppb.

9. **Minor comments**
   **Page 2, line 25: "positive radiative forcing on climate forcing" is better as "positive radiative forcing".**

We have modified this as suggested.

10. **Page 2, line 36: There has been a decrease in extreme ozone events, but the overall trend in ozone exposure is less clear, and recent work indicates a modest rise in ozone-related mortality (Sicard et al., 2021).**

We agree that the overall picture is more complex and that $O_3$ responses can vary across continental versus regional scales and for annual average versus episodic periods. We have added some up-to-date references to show the average change and trend in surface $O_3$ levels in continental regions with high emissions in the introductory section. See our response to reviewer 1 points 2 and 3.

11. **Page 2, lines 45 and 50: see the major comment above.**

Done.

**12. Page 4, line 101: Lin et al. (2020) showed that vegetation can also be a smaller sink for ozone in warmer climates due to reduced deposition. Is this effect also included in the study?**

Yes, in the UKESM1 changes in deposition rates of gas-phase species reflect underlying changes in vegetation and surface types simulated with the dynamic vegetation model JULES (Best et al., 2011; Clark et al., 2011) which is coupled to the UKESM1. Atmospheric concentrations and deposition rates thus respond to climate-driven changes in vegetation. We have now discussed this effect in section 4.2 where we discuss tropospheric $O_3$ changes:

Page 10, line 205:
*"Changes in $O_3$ dry deposition rates principally reflect changes in surface $O_3$ concentrations, although high temperatures under a warmer climate may reduce $O_3$ deposition rates due to vegetation stress (Lin et al., 2020)."*

**13. Page 9, line 160 and page 16, line 256, and in the abstract: Do increases in BVOC (mostly isoprene) really offset decreases an anthropogenic and biomass burning VOC? Butler et al., (2018) showed that biogenic VOC produce more ozone than anthropogenic VOC over these regions, so wouldn't an equal increase in BVOC at the expense of other VOC lead to an increase in ozone?**

We thank the reviewer for this point. We use 'offset' to indicate that there is a balance between these competing effects, but we agree that they are not equal, and have now revised this to use 'partly offset'. The reviewer is correct to point out that $O_3$ may increase under an increase in BVOCs at the expense of other VOCs. We have not assessed the individual impacts of anthropogenic and biogenic VOCs on $O_3$ changes under changing emissions and climate in this study, which are likely to be highly variable in space and time, but it would be interesting to quantify these influences in future. We provide a general assessment of $O_3$ changes and $O_3$ sensitivity to climate and emission changes. We have now included some discussions of the impacts of climate change on temperature-sensitive BVOC emissions and their subsequent impact on $O_3$ concentrations in the paper:

Page 12, line 232:
*"Surface $O_3$ concentrations are also influenced by climate change, reflecting changing natural emissions, $O_3$ production and destruction rates and $O_3$ deposition rates. …… but in continental areas these effects may be offset by $O_3$ increases due to higher soil $NO_x$ (Romer et al., 2018) and BVOC emissions, and by decreased $O_3$ deposition rates (Lin et al., 2020)……*

**14. Page 14, line 232: Please provide more information about how the "dominant" source region was identified in each part of the NOx-VOC space.**

We have modified the statement here to be clearer about how the dominant region was defined:

Page 16, line 274:
*"We determine the dominant region contributing to each bin in NO$_x$-VOC space based on the region contributing the greatest number of model grid cells to that bin."*

**15. Page 16, line 258 and page 18, line 267: How can changes in methane cause large changes in ozone itself, but not in ozone sensitivity to NOx and VOC? Methane is after all another VOC. This needs more disccusion.**

Oxidation of CH$_4$ makes a substantial contribution to surface ozone levels but it is a relatively uniform effect across the globe as it is governed by production in the free troposphere. Therefore, differences in regional O$_3$ sensitivity as shown in Fig. 7 are mainly driven by other O$_3$ precursors. We find that there is little change in O$_3$ sensitivity based on NO$_x$/VOCs when CH$_4$ concentrations are reduced greatly, as shown in Fig. 3 below. Hence the O$_3$ sensitivity is not greatly influenced by changing CH$_4$ concentrations.

[Figure]

**Figure 3.** Surface O$_3$ mixing ratios as a function of monthly mean NO$_x$ and VOC mixing ratios under **(a)** SSP3-7.0 and **(b)** SSP3-7.0-lowCH4. Monthly mean data for all months under SSP3-7.0 and SSP3-7.0-lowCH4 are used separately. The straight lines show the approximate thresholds of NO$_x$/VOC to distinguish VOC-limited (above the line) and NO$_x$-limited (below the line).

**16. Page 19, line 283, and in the abstract: Further emission controls on VOC and CH4 are unlikely to lead to reductions in winter ozone in these regions, as local photochemical production is extremely slow during winter. Reduced NOx emissions in winter act to reduce the titration of background ozone, so once the titration effect has been removed, what remains is the background ozone.**

We agree that NO titration effects are strong and O$_3$ production is slow in winter. However, these

are local effects depending on the local chemical environment, and background $O_3$ levels on a regional scale can still be reduced by reducing $O_3$ precursors emissions, even though this process is slow and occurs over a wider region. Therefore, emission controls on VOC and $CH_4$ are necessary over a large scale, and are useful in preventing regional $O_3$ pollution even in winter.

---

## Author Response (AR2)

Dear editor:

We thank you for your comments.

**"Reviewer 1 had commented on the usage of the terms 'concentration' and 'mixing ratio'. In the revised version, the authors have replaced some occurrences of 'concentration' by 'mixing ratio', but still on some occasions 'concentrations' is used inappropriately (e.g. page 3 at end of Introduction, page 4 when referring to Table 2, page 7; list not complete). Please revise the manuscript once more, paying close attention to this aspect."**

We have taken your concerns into account. We have replaced all incidences of 'concentration' with 'mixing ratio' for accuracy especially when expressing the unit of ppb, and considered 'concentration' on a case by case basis.